# The Head-to-Toe Hormone: Leptin as an Extensive Modulator of Physiologic Systems

**DOI:** 10.3390/ijms23105439

**Published:** 2022-05-13

**Authors:** Monica Misch, Prasanth Puthanveetil

**Affiliations:** 1Chicago College of Osteopathic Medicine, Midwestern University, Downers Grove, IL 60515, USA; monica.misch@midwestern.edu; 2Department of Pharmacology, College of Graduate Studies, Midwestern University, Downers Grove, IL 60515, USA

**Keywords:** leptin, metabolism, systemic role, physiological functions, leptin receptor

## Abstract

Leptin is a well-known hunger-sensing peptide hormone. The role of leptin in weight gain and metabolic homeostasis has been explored for the past two decades. In this review, we have tried to shed light upon the impact of leptin signaling on health and diseases. At low or moderate levels, this peptide hormone supports physiological roles, but at chronically higher doses exhibits detrimental effects on various systems. The untoward effects we observe with chronically higher levels of leptin are due to their receptor-mediated effect or due to leptin resistance and are not well studied. This review will help us in understanding the non-anorexic roles of leptin, including their contribution to the metabolism of various systems and inflammation. We will be able to get an alternative perspective regarding the physiological and pathological roles of this mysterious peptide hormone.

## 1. Introduction

The discovery of leptin arose simply from suspicion. Researchers noted than an isolated mutant mice colony lacking the *ob* gene possessed abnormal characteristics, such as hyperphagia, decreased energy expenditure, and early-onset obesity [1]. In 1953, a theory proposed the existence of a circulating molecule secreted by adipose tissue; this molecule was in direct communication with the hypothalamus and affects food intake, body weight, and long-term energy balance [2]. It was not until forty years later that this speculated molecule was characterized and identified. Derived from the mRNA transcript of the *ob* gene, this peptide hormone was comprised of 167 amino acids and named “leptin”, from the Greek work “lepto”, meaning “thin” [3]. Since its discovery, leptin’s pleiotropic effects have been found to influence hematopoiesis, angiogenesis, blood pressure, bone mass, and T-lymphocyte function, among other things [1,3]. The perception of leptin just as a peptide hormone that regulates body weight has evolved to now being seen as a signaling molecule capable of regulating physiological homeostasis [1,3].

## 2. Origins and Expression

In humans, the *ob* gene is expressed primarily in adipocytes; thus, serum leptin concentration is highly correlated to overall fat content in infants, children, and adults [4,5]. Leptin expression was found to be nearly double in subcutaneous fat relative to visceral and omental fat, and this principle has been proven true for both lean and obese individuals [4,5]. Leptin circulates either freely or bound to the extracellular portion of its receptor [5,6,7,8,9,10,11,12]. Binding of leptin on to its receptor allows it to cross the blood–brain barrier and admit access to the central nervous system. Free leptin has also demonstrated to have high blood–brain permeability, a process facilitated by tanycytes under conditions of capillary leaks and during debilitated blood–brain barrier integrity [5,6,7,8,9,10,11,12].

Although six isoforms of the leptin receptor have been identified, the primary receptor is the long form and resides in the arcuate nucleus of the hypothalamus [10,13,14,15,16]. If the neurons housing this receptor bind leptin, the receptor dimerizes and initiates a signaling pathway via cytoplasmic tyrosine kinase such as janus kinase (JAK) [10,13,14,15,16]. Following phosphorylation of the intracellular region of the receptor, the STAT proteins (STAT2 or STAT3) house Src-domains that allow them to anchor to the receptor [9,10,12,13,14,15,16,17,18,19]. Once activated, the STAT protein travels to the nucleus to initiate transcription of the pro-opiomelanocortin (POMC) gene; POMC has been identified as an anorexigenic peptide [20,21,22]. In addition, leptin inhibits neurons expressing the antagonist for POMC and neuropeptide Y (NPY) [20,21,22,23,24,25]. A lepti- mediated decrease in NPY also contributes to an anorexic effect, as NPY is a potent centrally acting orexigenic agent [20,21,22,23,24,25]. Fluctuations in leptin levels during fasting or starvation is a crucial factor [20,22,23,24,25]. In a normal, “fed” state, leptin levels are proportional to the amount or mass of adipose tissue in the body [20,22,23,24,25]. Aside from an individual’s baseline leptin levels, serum leptin can increase as much as 40% following an episode of overeating or decrease by 60 to 70% following chronic fasting [26,27,28]. Clearly, leptin is in direct communication with the central nervous system to flag acute changes in energy intake [26,27,28].

In addition to the CNS regions, there is also distribution of leptin receptors in the peripheral tissues [29]. This review also sheds light upon the role of leptin receptors across the physiological system, and the role of their accompanied signaling in regulating physiological functions.

## 3. Gastrointestinal System

The outreach of leptin expression extends into the gastrointestinal system and is continuous in the stomach. Leptin expression was greatest in the fundic region, where chief cells and parietal cells exhibit high reactivity to the hormone [30,31,32]. It should be noted that leptin receptors were detected in both healthy gastric epithelium and cancerous gastric cells [30,31,32]. With respect to physiological control of leptin secretion, high fat diets play an important role. Arita et al. [33] identified that a greater quantity of gastric leptin receptors became phosphorylated with consumption of a high fat diet [33]. The findings confirm that leptin secretion and leptin signaling are elevated on such a diet. Tracking down the GI tract, this report finds the link between leptin and intestinal health [33]. Mice lacking the leptin receptor were protected from intestinal dysbiosis and high-fat-diet-induced intestinal metaplasia, which reinforces the link between leptin secretion and gut health [33]. In the colon, a greater concentration of leptin receptors were found in ulcerative colitis patients despite serum leptin levels being normal [34,35]. It is worth considering in future research endeavors the density of leptin receptors in the GI tract, rather than just focusing on serum leptin levels. These findings characterize serum leptin as a mark of *localized* inflammation, especially in the gut.

## 4. Pancreas

There is an intimate and rather complex relationship between leptin and the secretory capacity of the pancreas. Leptin signaling induces the K+/ATP membrane receptor. The ultimate effect achieved is the conductance of K+ increases across the membrane, which hyperpolarizes islet cells to inhibit insulin secretion [36,37,38,39]. Leptin prevents the secretion of both insulin and glucagon [36,37,38,39]. While receptors for leptin are copious on β-islet cells and δ-islet cells, they are absent from the majority of glucagon-producing α-islet cells [36,37,38]. Thus, the inhibitory influence of leptin is more profound for insulin when compared to glucagon [39,40,41]. Please note that no profound effects on somatostatin secretion are observed following administration of leptin. Insulin signaling and leptin have got an interesting interaction that suggests a bi-directional feedback loop; as the secretion of insulin is known to stimulate leptin release, the released leptin limits insulin levels [39,40,41]. In this light, leptin can be considered as a regulating hormone for the pancreas. Another hypothesis for leptin’s inhibitory effect on insulin secretion is via the activation of intracellular cAMP signaling [42]. Leptin was shown to inhibit cAMP activity, and as a result, prevent the insulin secretory mechanism in the pancreatic cells [39,40,41]. An attributed mechanism is due to activation of phosphoinositide 3-kinase-cyclic nucleotide phosphodiesterase 3B (PDE3B) signaling nexus [43,44]. Activated PDE3 B could lead to enhanced breakdown of cAMP, mitigating the signaling responsible for pancreatic insulin secretion [43,44]. An additional possibility to consider is the toxicity of inflammatory cytokines and adipokines, which are induced or triggered by leptin and could act on β-cells to further compromise the ability to effectively secrete insulin [43,44]. Perhaps future investigation of the direct and indirect effects of leptin signaling on pancreatic islets would provide clarity.

## 5. Hepatic Tissue

Within the liver, leptin is considered an anti-steatosis hormone, but the levels and duration matters. The signaling cascade that leptin elicits in the liver targets a specific transcription factor that is a key component in lipid synthesis; its modulation by leptin mobilizes lipids [45]. The protective properties against lipid accumulation within the liver can be illustrated by observations made in models devoid of leptin receptors and accompanied by increased liver triglycerides and increased lipid deposition [46]. Moreover, leptin-deficient animal models saw a delay in liver regeneration, hypothesized to be a consequence of impaired angiogenesis and glucose transport to hepatocytes [47]. Although increases in circulating leptin would be presumably helpful, there seems to be a threshold that exists where chronic elevations exacerbate inflammatory and fibrogenic processes in the liver [45]. In fact, leptin was required for fibrosis to develop in mouse models with chronic liver injury [48]. Provided that the leptin receptor is expressed by Kupffer cells, hepatic stellate cells, and sinusoidal endothelial cells, there seems to be a direct mechanism of action for the development of fibrosis [45]. Likewise, the leptin-dependent induction of fatty acid oxidation and mitochondrial respiration taking place in the liver may induce oxidative stress [45]. When this fragile balance between leptin signaling and lipid mobilization becomes dysfunctional, individuals may be susceptible to pathological conditions. A positive correlation has been established between elevations in circulating leptin in non-alcoholic fatty liver disease patients, steatosis patients, as well as non-alcoholic steatohepatitis patients (NASH) [49]. It is important to consider the levels of leptin were influenced by factors such as age, gender, co-existing metabolic diseases, and percentage of body fat. Although increased leptin is not a direct indicator of these pathologies, it raises curiosity about its role as an indicator of ongoing metabolic complications.

## 6. Connective Tissue

In skeletal system, leptin is known to be a potent inhibitor of bone formation [50,51]. Bone loss becomes a concern for those with substantial fat stores, or in other words, subjects who present with an elevated body mass index, since there is a greater availability of fat cells to synthesize the hormone leptin. Leptin is also a potent inhibitor of bone formation [50,51]. Bone loss becomes a concern for those with substantial fat stores, elevated body mass index, and especially subjects who are insulin-resistant or have type 2 diabetes. A study evaluated the long duration effects of leptin by administering recombinant adeno-associated virus-rat leptin (rAAV-Lep) into the third ventricle of the hypothalamic region to understand the impact on weight gain and bone metabolism using female Sprague-Dawley rats which had sufficient leptin levels [52]. Interestingly in this study, at 5 weeks after vector administration, rAAV-Lep-administered rats developed lower cancellous bone volume and bone marrow adiposity. With the increase in duration of treatment, no significance differences were noted in cancellous bone but a major impact on bone adiposity and associated weight gain [52] was seen. Another study demonstrated that intracerebroventricular (ICV) administration of leptin reduces trabecular bone volume, most notably in the vertebral column [53,54,55]. It is unclear whether this loss of bone volume is due to the direct action of leptin on bone or its influence on the sympathetic nervous system (SNS). We also have found reports that demonstrate contradictory effects. For instance, leptin binding to its receptors on osteoclasts and osteoblasts elicits synthesis of the bone matrix. Leptin promoted the differentiation of osteoblasts, synthesis of Type I collagen, and allocation of osteocalcin [56]. Leptin by itself and in the presence of cytokines has been demonstrated to enhance collagenolytic and gelatinolytic properties in bovine cartilage explant cultures [57]. Leptin brings about this effect through the simultaneous activation of multiple matrix metalloproteinases (MMP1 and MMP13) and also involving transcription factors such as signal transducers and activation of transcription of family members (STATs) [57]. These effects were nullified by using an anti-leptin antibody [57]. Leptin inhibited osteoclast formation by increasing concentrations of osteoprotegerin, a protein that inhibits maturation of osteoclasts [58]. Aside from acting peripherally on bone, leptin’s activation of the ventromedial hypothalamus may have an indirect consequence of activating noradrenergic signaling at the osteoblasts, mediating its ability to impact bone mass [53,54,55]. It is difficult to isolate the effects of leptin on the skeletal system when leptin also acts alongside hypothalamic effectors, such as cortisol, IGF-1, estrogen, and parathyroid hormone [59]. Thus, the link between ICV administration of leptin and reduced trabecular bone sparks inquisition. Given that leptin has pro-osteogenic properties alone, there must be a complicated pathway that creates the overall consequence of reduced bone volume in hyperleptinemic cases that needs further evaluation.

## 7. Circulatory System

Once activated, T cells express a leptin receptor on their membrane, and when the receptor is revealed, the T cells become sensitive to changes in insulin concentration and nutrient availability [52,53]. Given that T-cell activation is energetically expensive, levels of serum leptin reflect nutrient availability for the process to begin [52,53]. The signaling pathways found to be mediated by the leptin receptor in T cells include upregulation of glucose intake, optimization of lactate production, proliferation, and production of inflammatory cytokines [52,53,54,55,56,57]. Additionally, the ability of T-cells to secrete inflammatory cytokines IL-2 and IFN-γ was found to be dependent on leptin availability [54,55,56,57]. Although leptin may be necessary for mounting a typical immune response, elevated serum leptin is a high-risk factor for many hematopoietic malignancies [58,59,60,61,62]. This was found to be true when leptin receptors were absent in normal promyelocytes, but leukemic promyelocytes housed mRNA of multiple isoforms of the leptin receptor [58,59,60,61,62]. The cascade of STAT3 and ERK 1/2 signaling that follows leptin receptor activation resulted in increased colony-forming ability, proliferation, and anti-apoptotic properties of human erythroleukemic cell lines. These findings are significant because they illustrate the direct effect of leptin on the pathological progression of hematopoietic malignancies [60,61,63]. A crucial note to make is that for certain blood cancers, serum leptin was elevated independent of the patient’s BMI [60,64], persuading against confounding factors such as obesity. These findings suggest focusing on the dysregulation of leptin itself, or leptin-mediated pathways as treatment for certain leukemias rather than focusing on BMI. To further highlight leptin’s role in the development of leukemia, leptin receptor mRNA was constitutively expressed in acute myelogenous leukemia and acute lymphocytic leukemia; leptin receptor expression also correlated well with immature CD34+ hematopoietic progenitor cells [59,60,65]. The above-mentioned findings express the importance of leptin during the blast or proliferative stage of blood cancers, rather than in more chronic stages. Early intervention and modulation of leptin signaling has the potential to be a promising route for leukemia research.

More recent studies in healthy patients point to leptin regulating blood flow and have identified a saturable, designated binding site for leptin on red blood cells [66]. The researchers found that leptin ultimately induced an increase in red blood cell-derived ATP, a recognized stimulus of blood flow [66]. Furthermore, leptin is also known to cause nitric oxide release and consequent vasodilation in endothelial cells [66]. Although leptin is known to play a crucial role in developing hematopoietic malignancies, it may also be beneficial to the circulatory system under normal physiological conditions.

## 8. Cardiovascular and Renal System

Leptin’s effects on the cardiovascular system are discordant and not well understood. Although both large population-based and clinical studies have found a positive correlation between hyperleptinemia and cardiovascular complications [67,68,69,70,71], it is unclear whether the adverse events are driven by hyperleptinemia alone. The accumulation of white adipose tissue that contributes to hyperleptinemia has other physiological consequences, such as obesity, hypertension, and diabetes and that could act as confounding factors for the cardiovascular events, as per the studies done in human subjects [67,68,69,70,71]. Regardless, it is clear that leptin has the potential to play a role in cardiovascular health. Leptin receptors have been located on hemopoietic cells [61,72,73,74,75,76], rightfully characterizing leptin as a systemic signaling molecule. Previous experiments have identified leptin as a promoter of platelet aggregation as well as an accelerator for wound repair, as per the observations done in human subjects [77,78,79,80,81,82,83,84,85,86]. These findings are evidence for leptin’s ability to facilitate the onset of thrombotic events or stroke in human subjects, which contributes to the growing interest of its role in regulation of the circulatory system. Likewise, leptin receptors have been identified in human atherosclerosis [87,88,89], which highlights the role of leptin signaling in endothelial dysfunction [87,88,89]. Leptin signaling has also shown to contribute towards hypertension [90,91,92,93,94,95] in mice and rats, an effect mediated by angiotensin II [90,91,92,93,94,95].

The endothelial cells have been demonstrated to have substantial leptin receptor gene expression. With activation of the leptin receptor, a tyrosine-kinase-dependent pathway initiates angiogenic processes [96,97,98,99] in human and animal cell model systems. Interestingly, Kang et al. [100] found that atherosclerotic lesions in human subjects had a greater expression of the leptin receptor gene when compared to histologically normal endothelium [100]. Note that obesity is identified as a major risk factor for atherosclerosis [101]. It would be logical to consider that the excess adipose stores in an obese individual could contribute to hyperleptinemia. Increased leptin levels have shown positive correlation with increased blood viscosity and enhanced platelet count with fibrinogen expression and activity [102]. This could explain leptin’s role in aggravating atherosclerotic lesions. 

Furthermore, leptin has been demonstrated to enhance the sympathomimetic effect, thus raising peripheral blood pressure [92]. This is not a mere correlation but supported by evidence as lean individuals who received exogenous leptin exhibited hypertension [103]. This illuminates the hormone’s ability to contribute to cardiovascular health independent of other contributing factors. In contrast, leptin’s signaling pathways do not always result in adverse outcomes; elevated serum leptin was also linked to cardioprotection [104]. Evidence from clinical trial has shown that leptin concentration was inversely associated with left ventricular and left atrial masses [101]. Further investigation is encouraged to determine if these effects are occurring through a separate signaling pathway or a mediated by its own receptor isoforms. Although some discrepancy still exists, there is a consensus that both excessive leptin and leptin deficiency would have an impact on cardiovascular health.

As a large molecular weight protein, leptin can be problematic for renal filtration. Hyperleptinemia is associated with impaired kidney function, including increased excretion of urinary albumin and a reduced glomerular filtration rate [96,97,98,99,100] in patients with chronic kidney diseases (CKD). Leptin is considered a uremic toxin, as elevated levels are associated with glomerular mesangial cell hypertrophy, fusion of podocytes, reduced metabolic activity in the proximal convoluted tubule, and thickened basement membrane [96,97,98,99,100], as observed in CKD patients. These consequences contribute to albuminuria, glomerular sclerosis, and apoptosis of nephrons.

## 9. Nervous System

On a biochemical level, leptin triggers anorexigenic neurons in the hypothalamus to synthesize pro-opiomelanocortin (POMC) and cocaine and amphetamine-related transcript (CART), which are two polypeptides that are known to limit food intake and increase energy expenditure [105,106,107]. Leptin has the power to simultaneously inhibit orexigenic neurons from synthesizing agouti-related-peptide (AGRP) and neuropeptide Y (NPY), which have antagonistic effects on satiety and promote feeding behavior [102,108,109] in both rats and mice. In other areas of the brain, leptin influences the lateral hypothalamus to decrease the expression of orexins, or general neuropeptides involved in food regulation and stress. Leptin is also known to directly activate a transcription factor called Steroidogenic factor-1 (SF-1) on neurons of the ventromedial hypothalamus regions [110,111] in rodents. When mice with leptin receptors knock down in generated SF1 positive neurons, these mice gained weight due to the loss of restriction on weight gain by leptin receptors [110,111]. It is evident that leptin is at work in many areas of the brain to control orexigenic urges as well as energy use. Additionally, the ventral tegmental area of the brain contains neurons possessing the leptin receptor. Leptin signaling in this region is a well-validated pathway involved in suppression of hunger [110,111]. Leptin receptors are ubiquitously expressed across astrocytes and microglia as well, which are targets for pro-inflammatory signaling within the hypothalamus [105,106,112,113,114,115]. The exact role of astrocyte and microglial residing leptin receptor needs to be understood further.

## 10. Immune System

Leptin serves as a communication link between the metabolic and immune systems [107,116,117]. The formation of a sufficient line of defense against pathogens is a highly energy-dependent process [118,119,120,121]. Thus, the presence of leptin receptors on most immune cells represents a close interplay between the body’s metabolic status and its ability to mount an immune response. With respect to innate immunity, leptin increases the cytotoxicity of natural killer cells, as well as increases the activation of granulocytes, macrophages, and dendritic cells [122,123,124,125,126,127,128]. As for adaptive immunity, leptin limits the proliferation of regulatory T cells but increases the production of naïve T cells and B cells [53,129,130,131,132,133]. Although the exact mechanism is unclear, the presence of leptin is believed to influence cell survival, as exogenous leptin was found to delay apoptosis via intracellular JAK, NF-kB, and MAPK pathways [134,135,136]. Overall, leptin induces an inflammatory response via immune cell activation, inducing chemotaxis and the release of cytokines [53,129,130,131,132,133]. Likewise, leptin plays a role in immunity by maintaining the balance of Type I and Type II Helper T cells [137,138,139]. Without sufficient levels of leptin, the ability to create CD-4 cells is compromised [137,138,139].

## 11. Sexual Dimorphism and Leptin

### 11.1. Sex-Specific Effects of Leptin: On Females

Serum leptin levels rise and fall throughout a woman’s menstrual cycle. In fact, estrogens induce leptin release [140,141,142,143,144,145,146]. The rise of estrogen that peaks mid-cycle is accompanied by a mid-cycle peak in leptin [140,141,142,143,144,145,146]. No studies have been done to confirm an ovarian contribution to serum leptin, but it seems that leptin levels can be used as a direct measurement of ovarian follicular health and its ability to produce other hormones, such as progesterone and LH. Based on observation in human subjects, during the menses phase, the level of leptin is close to or slightly over 15 ng/mL. During the follicular phase, the concentration rises to be over 15 ng/mL but less than 20 ng/mL [147]. During the ovulatory phase, leptin levels peak along with estrogen levels and reach approximately mid 20s (>20 ng/mL) [147]. With the luteal phase, the peak starts to decline and reaches back to the level of 20 ng/mL [147].

Leptin’s interaction with progesterone and LH remains ambiguous. Although leptin and progesterone show similar patterns of serum fluctuations during a healthy menstrual cycle [140,141,142,143,144,145,146], there is no evidence of regulation at a pre or post-translational level. With respect to LH, leptin receptor activation induces the STAT3 signaling pathway. It is the STAT3 induction, rather than leptin itself, that is responsible for the LH surge [148,149,150]. Nonetheless, leptin contributes directly or indirectly to the regulation of the reproductive cycle. 

The female reproductive system is a hallmark example of the need for research surrounding leptin expression. Normally, mammary epithelial cells have moderate expression of the leptin receptor gene. However, carcinoma cells within mammary epithelium showed a significant increase in leptin receptor expression [147,151,152]. It is important to note that these cells produce leptin themselves, more so than the non-cancerous control. This points to an autocrine signaling mechanism that may contribute to the proliferation and metastasis in breast cancer populations [147,151,152]. Interestingly enough, the tumors did not metastasize if they lacked the leptin or the leptin receptor gene [147,151,152], which confirmed the hypothesis. More recent studies confirm the link between overexpression of leptin and its receptor in both primary and metastatic cancers [153,154,155,156,157]. These findings stress the importance of a healthy BMI and fat content in cancer prognosis. Likewise, it also illustrates how obesity can be a detrimental factor for patients diagnosed with cancer, due to metabolic effects and also leptin-mediated direct effects.

### 11.2. Sex-Specific Effects of Leptin: On Males

Even though leptin receptors have been identified in the testes [158,159,160,161,162,163], the effects of leptin on the male reproductive system are less explored. Recent studies in rat models have shown that leptin is a direct inhibitory signal for testicular steroidogenesis [164]. Associations between high BMI, hyperleptinemia, low serum testosterone, and impaired sperm motility have been identified but not confirmed [158,159,160,161,162,163]. Serum leptin concentration following fasting has been shown to be lower in males (approximately >6.5 ng/mL compared to over 15.0 ng/mL) in comparison to females, suggesting females have a higher potential to generate leptin from comparable fat mass [147,165]. Interestingly, there has been reports that for females, there is a permanent drop of total leptin below 20 ng/mL in the post-menopausal stage [147,165]. Even for males, during their active adult life (30–50 years), the levels of leptin have been reported to be just over 10 ng/mL, permanently dropping to a level of just above 6 ng/mL after 50 years of age [147,165].

As such, the inhibitory influence of adipocyte leptin on androgens raises concern for elevated BMI values and infertility. Additionally, the role of leptin receptor stimulation by leptin released from both testicular and extra testicular tissues has not been well studied and needs attention. Definitely, enhanced plasma leptin levels have been well associated with both prostate cancer and testicular cancer in males, and the leptin receptor is a known target for treating these cancers in the male population [166,167,168]. There is also evidence that leptin is not a robust biomarker in males in comparison to females with the same types of cancer [169]. In lung and hematological cancers, the leptin levels in females are shown to have over 30 ng/mL, and for gastrointestinal and genitourinary cancers, the levels are over 20 ng/mL, in comparison to males, where the levels are less than 10 ng/mL [169]. In comparison to healthy conditions, the plasma levels are still high in males following cancer [169]. These observations can help us in drawing conclusions that leptin could be considered as a marker for cancer in males and an even more robust marker in females.

## 12. Leptin and Systemic Health

### 12.1. Overall Systemic Metabolic Homeostasis

This section will highlight on the divergent effects of leptin, which may not fit into a single organ system. The identification of leptin as a key player in metabolic homeostasis is rooted in its systemic effects when one abstains from eating. When leptin levels fall as a result of a fasted state, there is a neuroendocrine shift that promotes increased appetite with a concomitant effect of decreased energy use [140,141,142,143,144,145,146,149]. This overall effect is achieved by reducing testosterone, TSH, and the loss of LH hormone cycle in females [146,165,166,167,168,169,170,171,172,173,174,175,176].

Furthermore, leptin can induce the expression of insulin-like growth factor binding protein [177,178,179,180]. The mechanism is described by Won et al. as being direct and indirect. Leptin can directly and indirectly stimulate the expression of IGF-1 and IGF-2, based on the evidence from reported studies [177] in a teleost fish model. As a result, there is an enhanced glucose uptake and glycogen synthesis across the periphery. In skeletal muscle, leptin signaling could initially cause an increase in lactate production, but in contrast, it is important to understand that chronic leptin could also decrease muscle triacylglycerol accumulation [181,182,183,184,185,186], as per the observations from porcine myoblasts and rat and mice skeletal muscle tissues. 

#### Leptin Has a Crucial Role in Carbohydrate Metabolism

With respect to cellular glucose uptake, leptin shares many of its intermediate signaling pathway with insulin. The overlap begins at the level of phosphatidylinositol-3 kinase, and both hormones initiate the process of GLUT4 expression in skeletal muscle [187]. Since the two hormones work together to produce similar effects, the isolated actions of leptin are an ongoing investigation and is really tough to dissect. In the presence of normal levels of insulin and glucagon, leptin treatment was found to increase the expression of GLUT4 transporters up to two-fold [188]. These findings suggest leptin is an enhancer of glucose uptake and insulin sensitivity in skeletal muscle and also a negative regulator for GLUT4 recruitment, TBC1D1 and TBC1D4 [189], facilitating these effects. It is important to note that when insulin was removed as a confounding variable, leptin was not able to upregulate the insulin-stimulated uptake of 2-deoxyglucose or glycogen synthesis [190]. It appears that insulin has a permissive effect on leptin, and leptin cannot achieve the physiological effects mediated by insulin in the absence of an active insulin signaling. The combination of insulin and leptin can increase glucose oxidation up to six-fold compared to a control or an unstimulated state, whereas either hormone on its own displayed comparable increases in glucose decarboxylation reactions [190]. Leptin’s target(s) in the carbohydrate metabolism pathways are unclear, but previous research reports have offered some insight. Leptin was found to increase pyruvate dehydrogenase activity and activity of the Krebs cycle. These findings were significant and even higher than that observed with insulin *per se* [190]. The aforementioned evidence validates a crucial role of leptin in systemic carbohydrate metabolism, but the extent to which it is insulin-independent is unknown and needs more clarity.

In the liver, leptin has been shown to have a negative effect on gluconeogenesis [191,192]. Also, synthesis of cholesterol and bile acids are also known to be modulated by leptin either directly or through the central effects [187,190,193,194]. Acute and chronic leptin has been shown to have differential effects on fatty acid uptake and utilization [188,189,195,196,197]. Due to the ability in regulating glucose, fatty acids, cholesterol, and bile acids, leptin is considered as a crucial regulator of metabolic and systemic health.

### 12.2. Leptin Imbalance and Associated Diseases

A reduction in adipose tissue mass is inevitable when daily energy expenditure exceeds energy intake. When one’s adipose tissue mass falls below a certain threshold and leptin levels are consequently decreased, dysregulation of the HPA axis will ensue [198]. The cessation of menstrual periods, along with an elevated risk of osteoporosis, is linked to hypoleptinemia [199,200,201,202,203]. Although it is not confirmed that leptin is the sole contributor to these manifestations of decreased female hormone levels, it is considered a necessary factor [199,200,201,202,203]. Leptin replacement therapy was successful in restoring healthy menstruation cycles in those with adipose mass below an optimal threshold, signifying the importance of this hormone [202,204,205].

Congenital leptin deficiencies exist, even though it is rare [206]. Besides obesity and presence of excess adiposity, leptin deficiency also results in decreased insulin sensitivity, unfavorable lipid profile, and hepatic steatosis [9,173]. Exogenous leptin administration has been confirmed to improve all metabolic parameters and is the first line of treatment in these individuals [173].

In contrast, hyperlipidemia conditions are a growing concern for chronic myocardial health and homeostasis [101,104]. Elevated circulating leptin potentiates atherogenic factors, including inflammation, hypertrophy, platelet aggregation, proliferation of vascular smooth muscle, formation of reactive oxygen species, and endothelial cell dysfunction [77,78,79,80,81,84,191,194]. Hyperlipidemia goes hand in hand with obesity-related diseases, making it a major risk factor for atherosclerosis and heart disease [84,191,194]. Polyakova et al. confirmed prolonged hyperleptinemia led to an increase in blood pressure, heart rate, myocardial hypertrophy, systemic inflammation, and frequency of ischemic arrhythmias [207].

Given the anorexigenic effects of leptin in the brain, it has become a strong contender in the treatment of obesity. Despite efforts to reduce weight gain with exogenous leptin, there is failure to generate a physiological response in obese patients [208]. The term “leptin resistance” is used to explain the absence of expected physiological response to hyperleptinemia in these obese patients [208]. It suggests that hyperleptinemia may be a driving force for obesity, as chronic treatment with exogenous leptin that exceeds the individual’s required limit significantly increases body weight [208]. The nature of hyperlipidemia is also dependent on diet composition, which highlights the multifactorial aspects of metabolic management [209]. In mouse models, only a high-fat, high-sugar diet increased serum leptin values without a corresponding increase in NPY mRNA expression; thus, even though leptin was elevated, the mice remained hyperphagic [209]. This is a salient finding because it suggests both high-sugar and high-fat diets could be possible factors for leptin resistance. It opens the door for new research to find other possible factors that may introduce leptin resistance.

### 12.3. Genetic Predominance Affecting Leptin Resistance and Its Role in Obesity

Genetics play an important role in inducing obesity and leptin resistance [210,211,212]. Some of the common genes that contribute towards obesity and subsequently leptin resistance are: mutations in leptin (LEP), leptin receptor (LEPR), Melanocortin 4 receptor (MC4R), Proopiomelanocortin (POMC), Brain-derived neurotrophic factor (BDNF), Proprotein convertase subtilisin/kexin type 1 (PCSK1), and peroxisome proliferator-activated receptor (PPARs) [213,214,215,216,217,218,219]. Broadly, all these mutations have been associated with hyperphagia, metabolic dysregulation, and altered gut brain signaling, followed by weight gain and insulin resistance [210,211,212]. Excess circulatory leptin levels, along with defective leptin receptor signaling, could lead to leptin resistance, which further aggravates obesity, allowing for the initiation of a vicious positive feedback loop [213,214,215,216,217,218,219]. Roughly around eight different mutations have been reported in the leptin gene and with leptin receptors, and few single nucleotide polymorphisms have been reported either in cytokine homology domain or in their fibronectin type 3 domain [210,211,212]. MC4R acts as a major mediator in CNS for the anorexic effect of leptin [213,214,215,216,217,218,219,220,221,222]. To date, over 370 single nucleotide variations have been reported for MC4R, and among these over 65 variations have been predicted to be highly pathogenic in clinical subjects [213,214,215,216,217,218,219]. Even though not frequent, the monogenic mutations form the predominant genetic reason for causing obesity and leptin resistance in early childhood [220,221,222], which contributes to childhood obesity. Most of the above-mentioned gene mutations are known to influence leptin and its associated receptor signaling leading to pathogenesis of childhood obesity with severe metabolic complications [210,211,212]. A detailed understanding of mutations in these targets could help in alleviating childhood obesity.

## 13. Leptin as a Diagnostic and Therapeutic Tool

### 13.1. Diagnostic Tool

Fluctuations in the expression of leptin and its receptor in various disease conditions raises the possibility of its potential as a diagnostic biomarker. One example is using serum leptin as an additional anthropometric index to classify obesity. A study has revealed that elevated levels of serum leptin were positively correlated to standard markers of obesity and showed the strongest correlation with hip circumference [210]. Currently, body mass index remains the standard for classifying individual obesity, but this value has been identified as an imperfect representation of fat mass [211]. Routine measurements with serum leptin concentration may provide a more accurate depiction of individual fat mass, as long as further studies establish appropriate cut-off points for normal, overweight, and obese patients. Another example of the use of leptin as a biomarker can be found in dermatology. Significant deviations from normal serum leptin concentration are currently being investigated in psoriasis. Elevations in serum leptin are being used as a biomarker for both the diagnosis and severity of psoriasis [212,213]. One must consider serum leptin is not sufficient to make a diagnosis alone, but at the same time, its use as a diagnostic marker may aid physicians in solidifying a differential diagnosis.

Further, research is underway to investigate the use of leptin as a biomarker of malignancy. Serum leptin has been found to be significantly elevated in cases of prostate cancer and breast cancer, independent of obesity [214,215]. Surpassing the mere detection of cancer, leptin expression was significantly correlated to the stage of metastasis, as well as the degree of lymph node development [216]. With respect to colorectal cancer, immunohistochemical measures of leptin were used to accurately predict the cancer prognosis, independent of other indicators [216]. These findings were significant as it introduces leptin as a marker of clinical outcome. It is important to note that analysis of leptin can also be done using patient saliva. In the first study of its kind, researchers identified leptin as a preoperative indicator of parotid tumors; salivary leptin was used to distinguish tumor patients from healthy individuals [217]. The value of leptin as part of a cancer diagnostic workup is an interesting avenue to pursue.

Leptin may be used in conjunction with other hormones or cytokines to elevate its diagnostic value. For example, the ratio of leptin to adiponectin (or the inverse) is of interest [218]. These two hormones have contrasting effects on the manifestation of inflammatory processes, and thus the development of metabolic syndrome. Although metabolic syndrome has variable definitions, abdominal obesity is an obligatory component; this provides rationale for leptin to be evaluated as a diagnostic tool [218]. The leptin-adiponectin ratio (LAR) has been confirmed to be a better diagnostic marker for metabolic syndrome than either hormone on its own, that is elevated leptin levels or decreased adiponectin levels [218]. Additionally, the LAR were more correlated with current diagnostic values such as body mass index, body adiposity, and waist circumference in comparison to another marker for dysfunctional adipose, the visceral adiposity index [219]. A standardized reference range has yet to be set for LAR. Frühbeck et al. set a value that accurately accounted for cardiometabolic risk; patients with obesity, type II diabetes mellitus, and metabolic syndrome all had leptin-adiponectin relationships that met the criteria for increased risk [219]. Although the LAR is not ordinarily used, it may serve as an estimator that can potentially account for a larger number of identified subjects at risk than just considering leptin alone.

### 13.2. Therapeutic Tool

The form of leptin that is currently available for human therapy is recombinant methionyl leptin, or metreleptin. It has been approved by the Food and Drug Administration to treat congenital or acquired lipodystrophy, with the purpose of normalizing blood lipids [220]. The drug aims to reduce triglycerides and increase HDL and has been successful for leptin-deficient adults [220,221]. Congenital leptin deficiency is very rare, but leptin replacement therapy has been shown to also decrease body weight, total fat mass, food intake, and plasma insulin for this small cohort of individuals [221]. Additionally, leptin replacement therapy is being evaluated as a viable treatment option for hypothalamic amenorrhea. In these patients, their state of energy deprivation is characterized by reduced fat mass and thus serum leptin concentration. Exogenous leptin was found to resolve anovulation and normalize thyroid, adrenal, and gonadal axes in multiple drug trials [222,223]. For these reasons, leptin is a promising therapeutic agent in the realm of women’s health.

It is worth mentioning the efforts being made to find a use for leptin in the treatment of diabetes. Regarding type I diabetes, leptin administration was found to improve blood sugar levels, increase glucose uptake, and modulate the autoimmune destruction of pancreatic beta cells. Although persuasive, these findings were true for animal models but have yet to be replicated in clinical trials [224]. Conversely, clinical trials have been underway for type II diabetes. Therapeutic leptin did not elicit significant changes in body weight, body composition, or insulin sensitivity [225]. One must consider leptin resistance in these participants, as individuals were mostly overweight and obese. Perhaps recombinant leptin in non-obese individuals with type II diabetes would have a different outcomes, but future research is needed to confirm or deny these speculations.

## 14. Conclusions

To date, the influence of leptin and leptin receptor expression and regulation has been centered around obesity. Given that plasma leptin concentration is directly correlated to elevated body mass index and fat mass, it is rational to categorize leptin only as a weight-regulating peptide hormone [226]. In reality, the peripheral expression of leptin and its receptor may warrant extra obesogenic and anorexic effects but requiring thorough investigation [106,207,208]. The comorbidities associated with elevated fat mass may be partially explained from the peripheral pleiotropic effects of leptin [106,207,208]. The purpose of this review is to remove the fixation of leptin just as a target for only obesity research and instead to consider leptin as a connecting signal across multiple organ systems regulating metabolism, inflammation, and systemic homeostasis (Summary diagram–Figure 1). Our review portrays the diverse function of this peptide hormone in different organ systems. This review also reveals a gender-specific role for leptin with pronounced effects in females especially in the pre-menopausal stage or during the active reproductive cycle in comparison to age-matched male counterparts. Even though leptin levels can be considered as a systemic marker for obesity and metabolic syndrome in both genders, it is considered a more reliable diagnostic marker for different types of cancers in females.

As a clinical recommendation, leptin levels can be considered as an ideal diagnostic tool and marker for insulin resistance and metabolic syndrome in both genders, but leptin levels could serve as an appropriate marker for detecting cancer in females. Considering the high basal levels of serum levels in women during the pre-menopausal stage, it could play a regulatory role in systemic metabolic and endocrine functions. Thus, maintaining appropriate leptin levels in females could be quite crucial for their metabolic and systemic health. Many of the actions of leptin from head to toe in both genders are still unclear. Integrating information from basic and clinical studies should help us in revealing the unknown systemic role of this peptide hormone, both as a biomarker and as a therapeutic target.

## Figures and Tables

**Figure 1 ijms-23-05439-f001:**
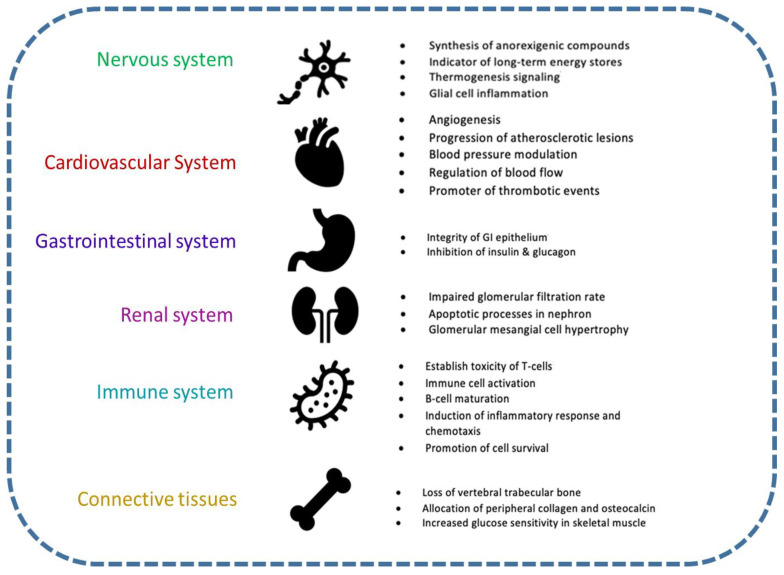
An illustration of the broad effects of leptin signaling within the human body. The role of leptin as an anorexic agent is well known. The role of leptin in regulating various physiological systems under normal and pathological conditions are explained here, including cardiovascular, gastro-intestinal, renal, immune, and skeletal systems.

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
