# Peer review of "The Head-to-Toe Hormone: Leptin as an Extensive Modulator of Physiologic Systems"

_ijms, 2022, doi:10.3390/ijms23105439_

Round 1
Reviewer 1 Report
The review is quite interesting and allows leptin to be considered as hormone that has numerous other effects on the body compared to only as a weight-regulating hormone. There are a lot of grammatical errors throughout the manuscript, and I recommended editing from a native English-speaking person. While body composition and diet are continually emphasised as factors influencing levels and sensitivity of leptin, there is an absence of any consideration to physical activity levels and more particularly exercise interventions. I would also encourage the authors to discuss the influence of muscle mass and myokines on leptin. Below are some other comments that need addressing.
Lines 12-15: I suggest mentioning the impact of leptin on health, because this is not clearly articulated.
Line 28: “It was not until…”
Line 44: Need to provide an example of why having high blood-brain permeability is of importance or worth noting.
Line 79: “ It is worth..”
Line 81: Please more clearly explain what is meant by “may be significant”? The point you are trying to make is difficult to understand.
Line 106: There should be a connect made with potential importance for people with insulin resistance and diabetes.
Line 165: How about the potential role for people with hypertension? People with greater blood viscosities and potentials for blood clot (e.g. fibrinogen)?
Line 171: “…contributing to hyperleptinemia..”
Line 198: “…pathway or mediated by…”
Line 218-220: Please revise this sentence. It is not clear how the mice gained weight?
Line 243: Is there are research surrounding leptin and treatment for cancer patients?
Line 261: “It is important..”
Line 268: “…fat content in cancer prognosis.:
Line 308: “Although it is not confirmed..”
Line 337: “…could be possible factors….”
Line 341: “To date…”
Line 343: “…it is rational..”
Line 347: “,,elevated fat mass…”
Author Response
Document attached with the responses.

Reviewer 2 Report
The authors submitted a narrative review with the aim of elucidating influence of leptin signaling through the mediation of its receptor on various physiological systems. The manuscript has logical structure, clear and legible figure. The physiological and pathological mechanisms by which leptin is involved in the regulation of metabolic processes, as well as its effect on target organs and tissues beyong its communication with appropriate receptors seem to be explained well. However, I would like to put forward several issues to discuss.
- The authors could extensively described the role of genetic predominance affecting leptin resistance and its role in obesity in special populations.
- The descriptions of the influence of leptin on CV, kidney, brain, WAT and muscles should be separately provided as animals as well as humans.
- There is no resoundingly clear report about diagnostic and predictive values of circulating leptin in males and females, whereas there is a large number of proof. Please, check and add more information.
- Please, re-write the section "Conclusion" taking into consideration aforementioned proposals.
- By far most powerful clinical recommendations for phisicians are required to be added in the section "Conclusion".
Author Response
Document attached with the response to reviewers.

Reviewer 3 Report
The paper entitled “The Head-to-Toe Hormone: Leptin as an Extensive Modulator of Physiologic Systems” includes relevant results/conclusions for the design and implementation innovative pharmacologic algorythms useful in the course of health service directed to management of overweight/obesity disorders accompanied by the systemic inflammation.
The Authors of this publication tend to suggest that their review research results can be helpful in understanding the non-anorexic roles of leptin including their contribution in metabolism of various systems and inflammation.
Remarks:
1.It is absolutely necessary to expand the scientific range/scope of the article by the addition of the following subsections:
- the effect of leptin on connective tissue;
- the effect of leptin on liver tissue;
- influence of leptin on carbohydrate metabolism;
- potential use of leptin (in medicine) as a diagnostic marker and as a therapeutic agent.
2.The resolution of Figure no. 1 is insufficient. Figure no. 1 need to be redrawn.
3.In addition, the articles used by the Authors – during the design and drawing of Figures no. 1 – must be mentioned/cited in the Figure no.1 description.
Author Response
Attachment with the response to the reviewer

Round 2
Reviewer 1 Report
There are so many grammatical errors in this manuscript. the writing is not concise and succinct. The logical for numerous arguments is confusing and structure of many sentences is very poor.
Reviewer 2 Report
The authors submitted a revised version of the paper along with comprehensive explanation of the ways by which the corrections were made. I am satisfied the revisions and I have no major flaws in the text, structure, scientific sound and tables/figures of the paper in its revised version. I feel the article became better than it was and the authors made serios efforts to improve it.
Reviewer 3 Report
The authors of the reviewed publication have modified the manuscript. They applied the necessary changes, adapting the manuscript to the recommendations of the reviewer.